# Some Novel Therapies in Parkinson’s Disease: A Promising Path Forward or Not Yet? A Systematic Review of the Literature

**DOI:** 10.3390/biomedicines12030549

**Published:** 2024-02-29

**Authors:** Anastasia Bougea

**Affiliations:** Department of Neurology, Aiginition Hospital, National and Kapodistrian University of Athens, 11528 Athens, Greece; annita139@yahoo.gr

**Keywords:** Parkinson’s disease (PD), gene therapy, cell-based therapies, targeted drug delivery, neuroprotective agents, magnetic field therapy

## Abstract

In light of the unsuccessful traditional therapies for Parkinson’s disease (PD) overmany years, there is an unmet need for the development of novel therapies to alleviate the symptoms of PD retardation or halt the progression of the disease itself. This systematic review aims to critically update some of the most promising novel treatments including gene therapy, cell-based therapies, targeted drug delivery, and neuroprotective agents, focusing on their challenges, limitations and future directions in PD research. Gene therapy in PD is encouraging, with AAV-based approaches targeting neurotrophic factors, dopamine production, and neuronal circuits in animal and clinical trials. A promising approach to targeted drug delivery for PD involves the use of nanotechnology to create drug delivery vehicles that can traverse the blood–brain barrier and deliver medications specifically to the regions of the brain affected by PD. Neuroprotective agents are compounds that have the ability to protect neurons from degeneration and death, and they hold great promise for the evolution of disease-modifying treatments for PD. Magnetic field therapy is a promising non-invasive method that promotes neural plasticity in PD. The establishment of standardized protocols for animal and human studies, safety, ethical considerations, and cost-effectiveness are the major challenges for the future research of novel PD therapies. The development of novel therapies for PD represents a promising path toward to effective personalized disease-modifying treatments for PD.

## 1. Introduction

Parkinson’s disease (PD) is the second most recurrent brain aging disease of an enormous personal and socioeconomic burden [median difference in annual costs 2261 dollars for the non-tremor type and -229 dollars for the tremor type] [1,2]. PD is signalized by the death of dopamine-producing neurons in the brain, leading to a wide range of motor (tremor, rigidity, bradykinesia, and postural instability) and non-motor symptoms such as dementia, sleep disorders, and autonomic dysfunction [3]. The progressive stages of PD are marked by a poor regulator of the motor complexities under levodopa per os/percutaneous treatments [4]. At this time, it is of vital importance to adopt invasive therapeutic options (device-aided therapies) [5]. However, these traditional therapies offer only asymptomatic relief without address the underlying neurodegenerative process. This deficiency could be attributed to the incapacity to moderate exogenous dopamine because of the denervation in the striatum, the brief half-term of levodopa, and the slowed gastric emission, which leads to an unpredictable fluctuation of plasma levodopa.

While traditional therapies have been the mainstay of PD treatment for many years, new and more effective treatment options are required for PD patients [6]. In recent years, novel therapies for PD have encompassed a wide range of approaches, including gene therapy, cell-based therapies, targeted drug delivery, neuroprotective agents, and magnetic field therapy [7,8,9,10,11,12].

The goal of these novel therapies is to alleviate the symptoms of PD as well as to downturn or suspend the PD course itself. This comprehensive systemized review aims to critically focus on some of the most promising novel therapies, discussing their challenges, limitations and future directions in PD research.

## 2. Materials and Methods

For the present review, the fundamentals of a systematic review [13] were limited to peer-reviewed articles with narrative evaluation [14]. I searched the PubMed/MEDLINE databases for peer-reviewed articles focusing on the role of novel PD therapies and their influence on PD pathogenesis, including papers published in the English language, with no time limit. The search was performed between September 2023 and November 2023. Ι used the terms “novel therapies”, “Gene Therapy”, “Cell-Based Therapies”, “Targeted Drug Delivery”, “Neuroprotective Agents”, “Adeno-associated virus (AAV) vectors”, “glial cell line-derived neurotrophic factor (GDNF)”, “aromatic L-amino acid decarboxylase (AADC)”, “CRISPR/Cas9”, “Stem Cell Transplantation”, “embryonic stem cells (ESCs)”, “neural stem cells (NSCs)”, “mesenchymal stem cells (MSCs)”, “MAO-B inhibitors”, “exenatide”, “Ambroxol”, “simvastatin”, “iron chelator”, and “Parkinson’s disease” with diverse combinations. After the searching test by title and abstract, only appropriate papers were read in detail. The snowballing process was performed to find additional articles. I principally analyzed studies on specific novel therapies for PD and their pathophysiological correlates.

### 2.1. Inclusion Criteria

Eligibility criteria were (1) studies on PD patients, (2) animal PD models, (3) cellculture studies, (4) publications in English.

### 2.2. Exclusion Criteria

The exclusion criteria were (1) Parkinsonian syndromes other than PD, (2) systematic reviews, letters to the editor, editorials, conference abstracts, and theses.

## 3. Results

From 20,114 papers, after eliminating 18,092 duplicates and rejecting particular studies according the exclusion criteria, 211 papers were reviewed based on the inclusion criteria. Finally, 108 studies were analyzed with regard to the novel therapies for PD (Figure 1).

Figure 2 shows an overview of the novel therapies for PD included in the present review. It summarizes the key points of the five novel therapies of PD included in this review—gene therapy, cell-based therapies, targeted drug delivery, neuroprotective agents, and magnetic field therapy—following a narrative analysis.

### 3.1. Gene Therapy

The concept of gene therapy involves introducing genetic material into targeted cells to correct or replace a defective gene, thereby restoring normal cellular function and providing therapeutic benefit [15,16] (Figure 2 shows a schematic presentation of this concept). In the context of PD, gene therapy aims to secure and regenerate dopaminergic neurons, ultimately delaying or stopping the evolution of the disease [16]. Several different approaches have been explored for gene therapy in PD, but for the scope of this review, I focus on gene delivery vectors, gene editing technologies, and gene transfer strategies [17].

#### Gene Delivery Vectors

Adeno-associated viruses (AAV) are part of a short (4.7 kB), non-enveloped, unpaired DNA genome of the parvovirus family, as one of the mostauspicious vectors systems for PD gene therapeutics [18]. AAV vectors have the ability to efficiently transduce neurons in the brain, making them well suited for delivering therapeutic genes to the affected areas in PD [19]. From a total number of 241 AAV vector-related clinical trials, 15 trials have been conducted including over 400 patients with PD [20]. Up to now, just three AAV-based therapies have been authorized in United States and Europe. Although promising, many studies were unable to achieve a cutoff at different stages of preclinical studies in PD animal models [21] and clinical trials [22,23]. The first limitation is the inappropriateness of patients as candidates to receive a specific AAV-based therapy, while a substantial patient sample was wrongly rejected. The second limitation is that the immune system of PD patients may react to the AAV infection and acquire a potent immunity hostile to the AAV capside by generating autoantibodies [24].

Among the widest studied gene cures for PD is the delivery of genes encoding for neurotrophic factors, the glial cell line-derived neurotrophic factor (GDNF), and neurturin (NRTN) [25]. Neurotrophic factors are essential for the viability and function of dopaminergic neurons; however, their encouraging delivery through AAV vectors was limited to only the preclinical stages of research. For example, a phase I trial conducted by Marks et al. [22,26] guaranteed the harmlessness of AAV-GDNF gene therapy in PD patients, showing an improved motor function and enhanced dopamine metabolism. However, this study, as with all earlier published studies, was unable to confirm clinical benefit [27,28,29]. The unclarified therapeutic mechanism of GDNF’s action could be responsible for the selection of inappropriate preclinical models (e.g., poor diffusion of delivery systems, downregulation of Nurr1) [30,31].

In addition to neurotrophic factors, other promising gene therapies aimed at restoring dopamine production or modulating the activity of neuronal circuits in preclinical models of PD. AAV transferring glutamic acid decarboxylase (AAV2-GAD) was delivered to the subthalamic nucleus with a notable advantage over sham surgery [32]. In addition, aromatic L-amino acid decarboxylase (AADC), a key enzyme in the dopamine synthesis, ameliorated motor function and reduced Parkinsonian symptoms in animal models [33]. Similar positive results were found with AAV-TH (tyrosine hydroxylase) or AAV-TH incorporated with AAV-AADC [34] and with AAV-TH incorporated with GTP cyclohydrolase I [35]. Furthermore, gene therapy approaches targeting specific neuronal circuits, such as the subthalamic nucleus or the globus pallidus, could also modulate dysregulated neural circuitry in PD [10]. By targeting striatal D1 medium spiny neurons (MSNs), Chen et al. [10] improved motor symptoms in both mouse and primate models of PD, suggesting the usefulness of chosen circuit regulation tools for PD therapy. While the preclinical and early clinical data for gene therapy in PD are encouraging, several issues should be resolved. One of the urgent challenges is the precise targeting and delivery of therapeutic genes to the affected areas such as substantia nigra and striatum [36,37,38]. The antegrade transfer of GDNF to the substantia nigra is not as powerful as operative repair in comparison with backward transfer [36,39,40]. There is also evidence of neuronal recovery with NRTN [38,41,42] and CDNF [43] after their striatal delivery of a rat 6-hydroxydopamine (6-OHDA) model of PD. Moreover, the blood–brain barrier (BBB) and the heterogeneous nature of PD pathology pose significant obstacles for effective gene delivery [44]. Therefore, it is necessary to further optimize of AAV vectors and gene delivery strategies to enhance their specificity and efficiency. Despite favorable preliminary data [22,26,32,34,35,45], the long-lasting effects of gene therapy on disease evolution and potential off-target effects must be carefully evaluated. Furthermore, the immune response to AAV vectors and the potential for insertional mutagenesis are important safety problems that should be resolved [24].

Despite these challenges, the area of a gene cure for PD is rapidly advancing, with innovative gene editing technologies. Clustered regularly interspaced short palindromic repeats (CRISPR)-Cas9 offers new opportunities for precise gene modification and targeted correction of genetic mutations associated with PD [46,47]. By using CRISPR-Cas9, coupled to the DNA-methyltransferase 3A catalytic domain (DNMT3A), Kantor et al. [48] decreased SNCA mRNA and protein in human induced pluripotent stem cell (hiPSC)-derived DNs from a PD patient with the triplication of the SNCA locus. According to the same method, Yoon et al. [49] managed the use of the CRISPR-Cas9 technique to delete the A53T-SNCA-specific PD gene in both in vitro and in vivo [49]. Moreover, Chen et al. [50] created SNCA−/− and SNCA+/− cell lines by deleting the endogenous SNCA gene in a clinical-grade human embryonic stem cell (hESC) line. Given that the cell replacement in PD patients has been proved susceptible to the host-to-graft transport of α-synuclein pathology, the created hESC lines transformed into dopaminergic neurons were disabled within the synthetic α-synuclein fibrils. These recombinant neurons were resistant to Lewy pathology, assisting the usage of the CRISPR/Cas9n systemin eliminating SNCA alleles averse to PD [50]. Apart from treating the symptoms of PD, this approach has the potential to address the underlying genetic causes of PD [11]. However, several limitations of CRISPR-Cas9 systems should be taken into in account such as Cas9 transport effectiveness into cells or tissue, off-target impact, and ethical problems regarding the human use of CRISPR technology [51].

### 3.2. Cell-Based Therapies

Cell-based therapies for PD are aimed at replacing the lost or damaged dopaminergic neurons in the brain, restoring the balance of neurotransmitters, and ultimately improving motor function and reducing symptoms [8]. Several types of cell-based approaches have been explored in preclinical and clinical studies, including fetal tissue transplantation, stem cell transplantation, and iPSC therapy [52] (Figure 2 Shows the key cell types).

#### 3.2.1. Fetal Tissue Transplantation

A well-studied and promising cell-based therapy for PD is the transplantation of dopamine-producing neurons derived from fetal tissue into the striatum of PD patients. This approach gained attention in the late 1980s and 1990s. Transplantation of dopamine neurons from the ventral midbrain (VM) of fetuses remain alive for a long time in the brains of PD patients showing physiological release of DA and symptomatic relief [53,54,55,56,57]. However, two double-blind placebo-controlled trials were unsuccessful inachieving their primary endpoints, while side effects as graft-induced dyskinesias (GIDs) were noted [58,59]. However, fetal tissue transplantation has ethical and logistical challenges, and the long-term efficacy and safety of this approach remains a topic of debate. As a result, researchers have been exploring alternative sources of dopamine-producing cells, such as iPSCs and embryonic stem cells.

#### 3.2.2. Stem Cell Transplantation

Stem cell transplantation involves the application of different stem cells—embryonic stem cells (ESCs), neural stem cells (NSCs), and mesenchymal stem cells (MSCs)—for replacing the lost dopaminergic neurons in the brain [60]. ESCs are potentially differentiable into any cell type in the body, including dopaminergic neurons, making them a promising candidate for cell replacement therapy. NSCs are specialized cells that can give rise to different types of neurons, while MSCs have been investigated for their potential in modulating the inflammatory response and promoting tissue repair in PD. In preclinical animal models, transplantation of fetal tissue, ESCs, NSCs, and MSCs ameliorated motor symptoms, increased dopamine levels, and promoted neuroprotection and neuroregeneration in the brain [61,62,63]. In clinical trials, some PD patients who received fetal tissue transplants or stem cell transplants have experienced better motor function and quality of life, although the long-term outcomes and risks of these treatments are still being evaluated [60].

#### 3.2.3. Induced Pluripotent Stem Cell (iPSC) Therapy

Induced pluripotent stem cells (iPSC), which are acquired from a patient’s own skin cells, hold great potential for the expansion of personalized cell-based medications for PD. By reprogramming a patient’s own cells into iPSCs and their differentiation into dopamine-producing neurons, researchers can potentially create a personalized source of cells for transplantation that could minimize the risk of rejection and other complications [64]. In a groundbreaking study published in 2017, researchers successfully generated iPSC-derived dopaminergic neurons from PD patients and transplanted them into the brains of non-human primates [65]. The transplanted neurons survived, integrated into the host brain, and exhibited functional recovery, demonstrating the potential of iPSC therapy for personalized cell replacement in PD. Adverse events include surgical injury, phlebitis, and hematoma.

Clozapine-N-oxide inverted or intensified the motor symptoms and behaviors of a PD mouse model with grafted mDA cells, suggesting the possibility of modulating neuronal recovery and improving treatment efforts after cell transplantation [66]. In a global initiative to the establishment of the stem cell-derived neural transplantation treatment for PD, iG-Force-PD consists of a global consortium with Europe-, USA-, and Japan-funded research teams who independently work to drive hESC- or hiPSC-derived mDA neurons from bench to clinical practice [67]. Lately, another G-Force-PD group reported the viability of transplanting self-iPSC-derived mDA neurons in a patient with PD [68].

In addition to iPSCs, embryonic stem cells (ESCs) are also being studied as a probable origin of transplantable dopamine-producing neurons for PD treatment [69]. Human (hESC)-derived DA neurons can fully backtrack motor deficits in laboratory models of PD [70]. Kirkerby et al. [71] started the first clinical trial in patients with a modest stage of PD in Europe, adopting the newly arisen hESC-derived DAergic cell product STEM-PD. iPSCs are advantageous compared to ESCs, such as in self-renewal and multi-cell differentiation. Therefore, iPSCs may overcome the various ethical issues regarding the demolition of embryos and the immune-related obstacles (it is facile to generate patient HLA-matched iPSCs compared with ESCs).

### 3.3. Targeted Drug Delivery

Nanotechnology-based drug transfer approaches offer several advantages for focused drug transportation to the brain, such as improved bioavailability, enhanced retention in the brain, and reduced systemic toxicity [72]. Various types of nanoparticles, including polymeric nanoparticles, liposomes, solid lipid nanoparticles, dendrimers, and mesoporous silica nanoparticles, were involved in the targeted transfer of therapeutic agents for PD, as follows:

Polymeric nanoparticles are widely used for special drug conveyance to the brain due to their biocompatibility, versatility, and tunable properties. Poly (lactic-co-glycolic acid) (PLGA) nanoparticles, for example, have been extensively studied for the delivery of dopamine agonists, neurotrophic factors, and gene therapies for PD [73]. By reforming the surface area of PLGA nanoparticles with ligands that target the transferrin receptor or the insulin receptor, researchers have demonstrated enhanced BBB penetration and intracellular delivery of therapeutic agents to dopaminergic neurons in PD experimental models of reducing PD symptomatology [73,74,75].

Compared to polymeric nanoparticles, liposomes are composed of lipid bilayers that can encapsulate hydrophilic and hydrophobic drugs, making them more suitable for the transport of various therapeutic agents forPD. By modifying the surface of liposomes with ligands that target receptors or transporters across the BBB, brain uptake and the therapeutic efficacy of encapsulated drugs were enhanced in rat models of PD [76,77]. Furthermore, the ability to functionalize liposomes with cell-penetrating peptides, antibodies, or aptamers offers additional opportunities for specific drug delivery to certain cell types within the brain (Figure 2 shows a schematic diagram of the drug delivery-mediated nanoparticle system). Interestingly, Zheng et al. [78] used two aptamers, F5R1 and F5R2, to suppress the α-syn accumulation and focused the cellular α-synuclein to corruption in SK-N-SH human neuroblastoma cell line (SK-N-SH) cells and primary neurons. In this way, these aptamers blocked the α-syn binding to mitochondria and prevented the mitochondrial malfunction and cell deficiencies generated by α-syn overexpression.

Solid lipid nanoparticles (SLNs) and nanostructured lipid carriers (NLCs) are emerging as promising platforms for targeted drug shipment to the brain. SLNs are composed of biocompatible lipids that can encapsulate hydrophobic drugs, while NLCs combine solid lipids with liquid lipids to improve drug loading and release properties [79]. SLNs and NLCs have been investigated for the delivery of dopamine agonists, antioxidants, and anti-inflammatory agents for PD, showing enhanced brain accumulation and sustained release of therapeutic agents in preclinical studies [80,81]. The propensity to adjust the lipid composition and surface qualities of SLNs and NLCs offers opportunities for maximizing their brain-targeting capabilities and therapeutic efficacy for PD.

Dendrimers are highly branched macromolecules with precise 3D structure and minimal polydispersity that have gained attention for targeted drug delivery to the brain [82]. PAMAM (polyamidoamine) dendrimers, for instance, have been used to deliver neurotrophic factors, anti-inflammatory agents, and disease-modifying therapies for PD [83]. Recently, the complex of curcumin CUR-PAMAM DG4.5 inhibited oxidative stress activity and α-synuclein aggregation [84]. These results suggested that the PAMAM dendrimer may act not only as a delivery system but also as a nanodrug in compound with other bioactive molecules in the therapy of PD.

Mesoporous silica nanoparticles (MSNs) are a versatile class of nanoparticles with ordered pore structures that offer high drug loading capacities, tunable pore sizes, and excellent biocompatibility, making them attractive for the delivery of small molecules, proteins, and nucleic acids for PD [85]. Furthermore, the ability to tailor the pore structure and surface chemistry of MSNs offers opportunities for optimizing their brain-targeting capabilities and therapeutic efficacy [86]. The prosperous L-DOPA loading is distinctly hopefulfor more studies on in vitro and in vivo biocompatibility, in order to appraise the potential of clinical application [87].

Apart from drug delivery, there is an increasing interest in nanoparticles such as CeO_2_ as ROS scavengers in PD. These nanoparticles reduce the dose-dependent toxic effects of alpha-synuclein [88]. So far, little evidence exists on CeO_2_ NPs as a therapy tool for PD. CeO_2_ NPs maintained neuronal feasibility on a 1-methyl-4-phenyl-1,2,3,6-tetrahydropiridine (MPTP) mouse model of PD [89], Figure 3.

### 3.4. Neuroprotective Agents: What Is New in an Old Story?

In addition to approaches aimed at replacing or repairing damaged neurons, there is also a great deal of interest in the growth of neuroprotective agents that may slow or halt the progression of PD. Neuroprotective agents are compounds aiming to protect neurons from degeneration and death, as promising tools for the enlargement of disease-modifying medications for PD (Figure 2 summarizes the five types of neuroprotective agents that were analyzed here).

Since 1962, one of the most widely studied neuroprotective agents for PD is a MAO-B inhibitor such as selegiline and rasagiline. Selegiline is an antioxidative agent with antiapoptotic effects [90]. So far, the two large prospective, double-blind (DATATOP and SINDEPAR) (for the purpose of this review, I focused to the most important large studies) clinical trials showed significant reductions in on/off fluctuations and levodopa dose. Previous reviews also indicated the long-lasting preservation of the premier asset, proposing the neuroprotective impact of early-onset selegiline therapy [91,92,93]. To improve its bioavailability, innovative transfer types of selegiline include transdermal and per os decomposing tablets alleviating nicotine and cocaine dependence [90].

Rasagiline is a stronger MAO-B inhibitor without amphetamine dependence compared to selegiline. Rasagiline works by inhibiting the activity of enzymes that are involved in the breakdown of dopamine, thereby increasing the availability of dopamine in the brain and reducing oxidative stress and inflammation [94]. Rasagiline provides long-term motor improvement (OFF-time reduction), supporting stable outcomes due to neuroprotection or effects on compensatory mechanisms in early PD [95,96,97,98].

In the late stage, MAO-B, mainly located in glial cells, is crucial for dopamine metabolism in the brain. Cell and molecular studies revealed interesting properties of selegiline opening new possibilities for neuroprotective mechanisms and a disease-modifying effect of MAO-B inhibitors [90]. Upon activation, MAO-B inhibitors express multiple proinflammatory cytokines such as IL-6, IL-1β, and TNF-α as well as nitric oxide and reactive oxygen molecules [99]. In a rotenone-induced rat model of PD, rasagiline decreased TNF-α mRNA expression in brain homogenates [100]. Selegiline has proven anticancer–antiapoptotic activity in cell lines [90]. Selegiline also ameliorates the vascular function in patients with coronary heart disease, either with or without diabetes [101]. Importantly, the variety in pharmacological activities observed in vitro and in vivo, albeit still far from PD clinical relevance, highlights either unexplored mixed capacities of MAO enzyme protein or novel MAO inhibitor types [90]. More large prospective studies are needed to confirm these findings.

Drug repurposing is a new international plan empowered by international linked clinical trials to explore the potential of non-anti-Parkinsonian validated medications for PD [102,103,104]. Although an old antidiabetic drug, exenatide (long acting glucagon-like peptide-1 (GLP-1) receptor agonist) may deploy neuroprotective and anti-neurotoxic action both in PD mice models [105,106,107] and PD patients with a constant motor and cognitive improvement over a 12 month-follow up [108]. However, this positive outcome was not replicated for the NLY01-a brain-pervasive, pegylated, prolonged-term form of exenatide [109]. Ambroxol, an anti-mucosal drug, augmented glucocerebrosidase (GBA) activity, decreased α-synuclein, and phosphorylated α-synuclein concentrations in PD mouse models [110] and in PD patients (+/−mutant GBA) [111]. Despite the established effect of lowering blood cholesterol, simvastatin as a HMG-CoA reductase inhibitor failed to delay PD progression [112]. Preclinical results for ATH434, iron chelator, are promising in the iron accumulation in SN, and therefore alleviate motor deficit in PD mice [113], but more studies are warranted to validate this benefit in PD cohorts.

## 4. Magnetic Field Therapy

High-frequency repetitive transcranial magnetic stimulation (rTMS), as a non-invasive, safe PD therapy, improves motor symptoms by generating a magnetic field on the basal ganglia that leads to the neural plasticity of the activated neurons in the motor cortex [114,115,116] (Figure 2 shows the magnetic schematic generation in the motor cortex of a patient with PD). Importantly, the beneficial effect in the off state was persistent for an increased number of sessions [117]. In the same line, other motor symptom such as gait freezing wase ameliorated after rTMS sessions [118]. However, intermittent theta-burst stimulation (iTBS) of the motor and prefrontal cortices, as a novel type of rTMS, was unsuccessful improving the motor functionality of PD patients [119]. The inconsistency could be explained by the small sample size and the lack of uniform TMS parameters and well-designated studies.

Regarding the non-motor symptoms, 19 placebo-controlled studies of high TMSfrequency (5 Hz) were demonstrated to be beneficial for depression [120] by diminishing theregional cerebral blood flow in the right medial frontal gyrus in PD patients [121]. Interestingly, the high striatal excitability results in long-lasting sensitive depression to dopamine in PD rats [122]. By contrast, >10 Hz rTMS studies failed to enhance mood [123,124]. The brief TMS periods could partially justify the adverse outcomes. On the other hand, cognitive PD-related domains such as executive processing, solving ability, and visual attention were improved in 15 Hz TMS [125] but not in 5 Hz [120] studies on motor and dorsolateral prefrontal cortices of PD patients. Overall, no studies have reported serious adverse effects during stimulation.

## 5. Challenges, Limitations, and Future Directions

While the field of novel therapies for PD holds great promise, there are also significant challenges and unanswered questions that need to be addressed as these therapies progress towards clinical use.

One of the key challenges is the need to develop safe and effective delivery methods for the various novel therapies being investigated. PD research lacks clinically predictive animal models that conscientiously replicate all foremost features of parkinsonian neurodegeneration and disease progression [126]. For example, using CRISPR-Cas9 technology in PD models, knocking out PINK1 and Parkin did not reproduce the PD-like motor and behavioral profiles in patients with PD [127]. Furthermore, gene therapy and cell-based therapies require precise and targeted delivery strategies to ensure that therapeutic agents reach the specific areas of the brain where they are needed without causing unintended side effects.

One of the major challenges is to optimize the survival, integration, and functionality of transplanted cells in the host brain, as well as to minimize the likelihood of unfavorable effects, such as tumor growth or immune rejection [128]. Transplantation outside the basal ganglia was demonstrated as a cost-effective cell delivery method to remedy the motor capacity of PD patients, but more prospective studies are needed to confirm these results [129]. The ideal origin for transplantation is one of cells that could be competent and affordable in conjunction with a likely curative effectiveness after grafting [130]. The cells have to lose their proliferative properties after grafting and their infectious components, and be immunologically in accordance with the host [131]. In addition, there is a need for prospective analysis to enhance our comprehension of the long-term effects and potential risks correlated with novel therapies for PD. Ethical considerations (e.g., destruction of a human embryo), regulatory requirements, and the high cost of cell-based therapies also pose significant barriers to their widespread application [132]. Advances in cell reprogramming techniques, such as the utilization of gene editing technologies to upgrade the safety and efficacy of iPSCs, will also be critical for advancing the area of cell-based therapy for PD [133]. Additionally, the establishment of standardized protocols for cell production, transplantation, and monitoring will be essential for ensuring the reproducibility and reliability of cell-based therapies across different research and clinical settings.

While the potential of nanotechnology-based targeted drug delivery for PD is evident, several challenges prevent from translating these approaches into clinically viable PD therapies [134]. One of the key challenges is the scalability and reproducibility of producing nanoparticles with consistent physicochemical properties and biological performance. Manufacturing processes for nanoparticles, such as shape, size, charge, and moiety, need to be refined to ensure clinical-grade production while maintaining the integrity of the encapsulated therapeutic agents. Moreover, the long-term stability, storage conditions, shelf life, and eco-friendly nature of nanoparticles should be carefully evaluated to ensure their safety and efficacy as drug delivery systems for PD [135]. Another challenge is the potential toxicity and immunogenicity of nanoparticles, as well as their interplay with the biological ambiance [136]. Additionally, the clinical translation of nanotechnology-based targeted drug delivery for PD requires careful consideration of regulatory and ethical aspects, as well as the development of appropriate animal models and preclinical assays. The ability to accurately predict the pharmacokinetics, distribution, metabolism, and excretion of nanoparticles in humans is crucial for advancing these technologies into clinical trials and ultimately into the clinic. For example, novel, non-invasive, alternative methods for per os, transbuccal, or nasal nanocarrier drug delivery may remove the obstacle of patient compliance [137]. Moreover, the cost-effectiveness, scalability, and market acceptance of nanotechnology-based drug delivery systems need to be evaluated to ensure their commercial viability and accessibility to patients with PD [138].

In order to improve magnetic field techniques, there is a need to establish international cutoffs of TMS parameters. A uniform TMS study protocol should include the same number of TMS sessions, potential effects concomitant medications, homogeneous measurements of the same stimulation sites, and pulse counts [114]. Well-designated studies with long follow-up including a complete battery of clinical/neuropsychological and imaging tools are warranted to confirm the efficacy of magnetic neuromodulating therapies in PD.

Another challenge is the need to develop robust and validated biomarkers to assess the efficacy of the novel therapies for PD included in this review. Fluid and neuroimaging biomarkers that can accurately reflect the progression of PD and the effects of treatment are essential for the evolution of disease-modifying therapies that delay or stop the progress of the disease [139]. By harnessing the power of gene therapy, cell-based strategies, targeted drug delivery, and neuroprotective agents, researchers promote the novel époque of personalized and disease-modifying treatments for PD.

## 6. Conclusions

The expansion of novel therapeutics for PD represents a promising path forward that has the potential to transform the treatment of this devastating neurodegenerative disorder. Gene therapy encourages the transforming of the PD therapy by addressing the underlying molecular mechanisms of the disease. The preclinical and early clinical data for gene therapy in PD are encouraging, with promising results from AAV-based approaches targeting neurotrophic factors, dopamine production, and neuronal circuits. However, there are significant challenges and limitations that need to be overcome to entirely recognize the potential of gene therapy in PD. Further research and development efforts are needed to optimize gene delivery, enhance long-term safety and efficacy, and explore emerging gene editing technologies. PD-related celltherapies are not independent therapies but should always be advised together with validate regimens [8]. By leveraging cutting-edge nanotechnologies and innovative approaches, researchers are making strides towards the development of new treatments that could offer new hope and improved outcomes for patients with PD. While there are significant challenges that lie ahead, the progress being made in the field of novel therapies for PD is proof of the commitment and ingenuity of the scientific community. As we continue to promote our acknowledgment of underlying pathological pathways and the potential of novel therapies, we are moving closer towards a future in which effective disease-modifying treatments for PD may become a reality in clinical practice.

## Figures and Tables

**Figure 1 biomedicines-12-00549-f001:**
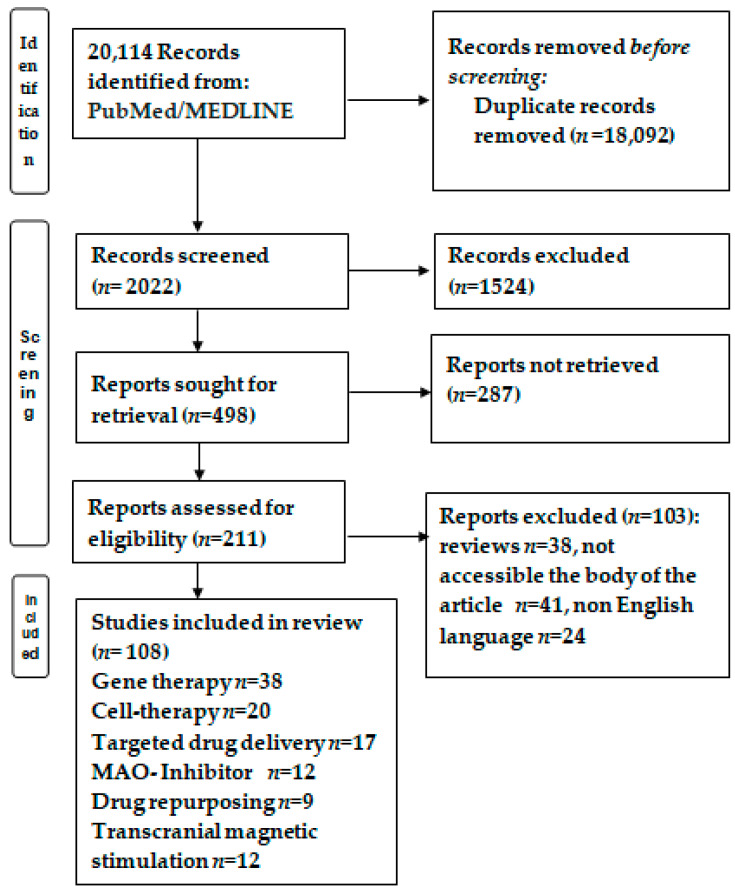
Flowchart of the study selection. After conducting PUBMED/MEDLINE searches and removing duplicates or documents that did not meet the inclusion criteria, 108 papers were deemed eligible for the narrative analysis.

**Figure 2 biomedicines-12-00549-f002:**
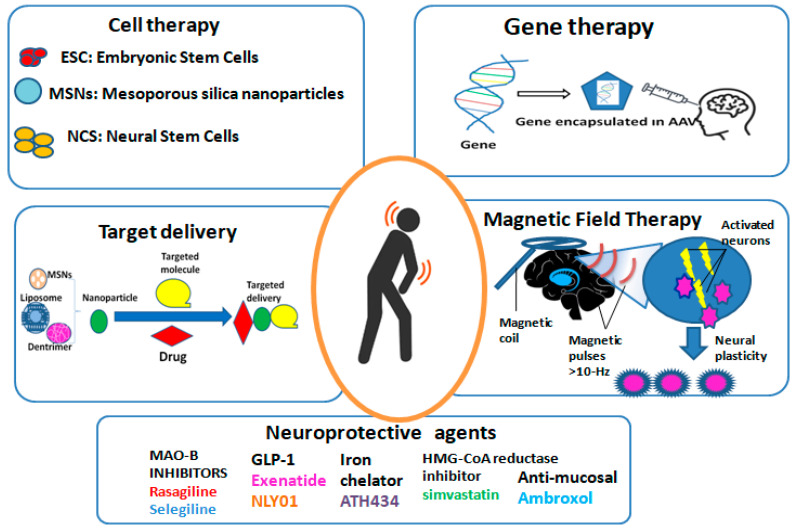
Overview of the novel therapies for PD. AAV: adeno-associated virus; ESC: embryonic stem Cells; MAO-B: type-B monoamine oxidase; MSNs: mesoporous silica nanoparticles; NCS: neural stem cells.

**Figure 3 biomedicines-12-00549-f003:**
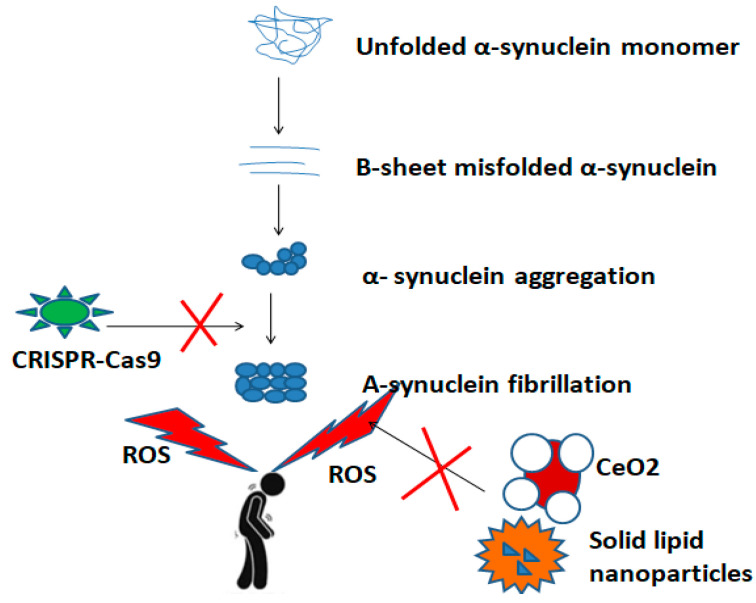
A schematic overview of inhibitory action of nanoparticles and gene editing technologies, such as the CRISPR-Cas9 system during the pathogenetic pathway of PD.

## Data Availability

Not applicable.

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
