# Peer review of "Some Novel Therapies in Parkinson’s Disease: A Promising Path Forward or Not Yet? A Systematic Review of the Literature"

_biomedicines, 2024, doi:10.3390/biomedicines12030549_

Round 1
Reviewer 1 Report
Comments and Suggestions for Authors
The review manuscript by Anastasia Bougea on Novel therapies in PD provides some useful information about novel basic and clinical studies. Although it is interesting, however, in my opinion, the review manuscript is not comprehensive. For example, in my previous comments, I had suggested the author to include other novel aspects in PD therapy (magnetic field therapy, drug repurposing) in addition to what had been covered in the manuscript. Since the author kept the previous structure of the manuscript with some revised text and content, I believe that it would be appropriate to change the first part of the title from "Novel Therapies in Parkinson’s Disease" to Some Novel Therapies in Parkinson’s Disease".
The drug repurposing is very active and dynamic area of research in PD (113 review article on the subject in the Pubmed) including PMIDs, 37328112, 33459662, 30066310, 34309236, and 33815053, as some examples.
Minor point:
-Please define G-Force-PD.
Author Response
Minor point:
-Please define G-Force-PD. MY RESPONSE: G-Force PD is a global consortium with Europe, USA, and Japan funded teams working on developing a stem cell-derived neural transplantation therapy for Parkinson’s disease (PD).
MY RESPONSE: After the extensive revisions, I have made a more comprehensive review by providing a systemized research of the literature . I focused from the first moment I ONLY focused on specific novel therapies in Parkinson’s Disease including gene therapy, cell-based therapies, targeted drug delivery, and neuroprotective agents. In the context of gene therapy I analyzed 3 important aspects critically on gene delivery vectors, gene editing technologies, and gene transfer strategies. In the context of cell-based therapies I analyzed 3 important aspects critically fetal tissue transplantation, stem cell transplantation, and iPSC therapy. In the context of targeted drug delivery I analyzed the polymeric nanoparticles, liposomes, solid lipid nanoparticles, dendrimers, and mesoporous silica nanoparticles.
In the context of neuroprotective agents I analyzed two old MAO-B inhibitors in order to critically explore a innovative potential. Here, it would be better to add the drug repurposing as old agents Exenatide, Ambroxol, Simvastatin and iron chelation in order to reveal a novel possibility of treatment of PD.
‘’Drug repurposing is a newly international plan empowered by the international Linked Clinical Trials to explore the potential of non- antiparkinsonian validated medications for PD[100-102]. Although old antidiabetic drug, exenatide (long acting glucagon-like peptide-1 (GLP-1) receptor agonist) may exert a neuroprotective and anti-neurotoxic action both in PD mice models[103-105] and PD patients with a constant motor and cognitive improvement over a 12 month-follow up[106]. However, this positive outcome was not replicated for NLY01-a brain- pervasive, pegylated, prolonged-term form of exenatide [107]. Ambroxol, an anti-mucosal drug, augment the glucocerebrosidase (GBA) activity and decrease α-synuclein and phosphorylated α-synuclein protein levels in PD mice models[108] and in PD patients (+/- mutant GBA)[109]. Despite the established effect in lowering blood cholesterol, simvastatin as a HMG-CoA reductase inhibitor failed to delay the PD progression [110]. Preclinical results for ATH434, iron chelator, are promising in the iron accumulation in SN, an therefore, alleviate motor deficit in PD mice[111], but more studies are warranted to validate this benefit in PD cohorts. ‘’ (lines380-394, 395-396 page 12,13) (for the purpose of this review I focused on specific studies on the ’Drug repurposing just to reveal their neuroprotective potential.
Therefore the figures 1 and 2 were updated properly
For all these reasons I agree with you to correct more precisely the title as’’ Selected Novel Therapies in Parkinson’s Disease: A Promising Path Forward or not yet? A Systemized Review of the Literature”

Reviewer 2 Report
Comments and Suggestions for Authors
The manuscript has been revised according to previous comments.
Author Response
ΜY RESPONSE : Thank you
